# Utilization of Lime Mud Waste from Paper Mills for Efficient Phosphorus Removal

**Hong Ha Thi Vu [1], Mohd Danish Khan [2], Ramakrishna Chilakala [3], Tuan Quang Lai [2], Thriveni Thenepalli [1], Ji Whan Ahn [1], Dong Un Park [4] and Jeongyun Kim [1,*]**

[1] Center for Carbon Mineralization, Mineral Resources Division, Korea Institute of Geoscience and Mineral Resources, 124 Gwahak-ro, Gajeong-dong, Yuseong-gu, Daejeon 34132, Korea; hongha@kigam.re.kr (H.H.T.V.); thenepallit@rediffmail.com (T.T.); ahnjw@kigam.re.kr (J.W.A.)

[2] Resources Recycling Department, University of Science and Technology, 217, Gajeong-ro, Yuseong-gu, Daejeon 34113, Korea; danish0417@ust.ac.kr (M.D.K.); tuanlai@ust.ac.kr (T.Q.L.)

[3] Department of Bio-Based Materials, School of Agriculture and Life Science, Chungnam National University Daejeon-34132, Korea; chilakala_ramakrishna@rediffmail.com

[4] Center for Climate Technology Cooperation, 17th Floor, Namsan Square Bldg., 173, Toegye-ro, Jung-gu, Seoul 04554, Korea; dongun.park@gtck.re.kr

* Correspondence: kooltz77@kigam.re.kr

**Abstract:** In this study, we utilized lime mud waste from paper mills to synthesize calcium hydroxide ($Ca(OH)_2$) nanoparticles (NPs) and investigate their application for the removal of phosphorus from aqueous solution. The NPs, composed of green portlandite with hexagonal shape, were successfully produced using a precipitation method at moderately high temperature. The crystal structure and characterization of the prepared $Ca(OH)_2$ nanoparticles were analyzed by field emission scanning electron microscopy, Fourier transform infrared spectroscopy, and X-ray diffraction. The effects of $Ca(OH)_2$ NP dosage and contact time on removal of phosphorus were also investigated. The results show that the green portlandite NPs can effectively remove phosphorus from aqueous solution. The phosphorus removal efficiencies within 10 min are 53%, 72%, 78%, 98%, and 100% with the different mass ratios of $Ca(OH)_2$ NPs/phosphorus (CNPs/P) of 2.2, 3.5, 4.4, 5.3, and 6.2, respectively. Due to the efficient phosphorus removal, the calcium hydroxide nanoparticles (CNPs) could be a potential candidate for this application in domestic or industrial wastewater treatment.

**Keywords:** calcium hydroxide; lime mud; nanoparticles; phosphorus removal

## 1. Introduction

Pulp and paper industries are producing an enormous amount of pulp annually to meet the ever-growing demand for papers and packaging materials [1–3]. Statistics also revealed that the global paper and cardboard production rose from 391.2 to 410.9 million metric tons from the year 2008 to 2016 [4]. Countries such as China, the United States, and Japan account for more than 50% of total paper and cardboard production. China alone produces more than 111.3 million metric tons (as of 2016) and currently is the largest producer of pulp and cardboard, followed by the United States, Japan, and South Korea [5]. However, organic and inorganic wastes (dregs and ash) are produced in huge quantities as byproducts, causing severe ecological and environmental issues. Lime mud is one such byproduct and its estimated outcome accounts for approximately 0.47 $m^3$ $tons^{-1}$ of pulp produced [6]. Lime mud is produced during the wood chips-to-pulp conversion process for paper production. The pulp is extracted from those wood chips through sodium hydroxide treatment and sodium carbonate is formed as a byproduct. For the recovery of sodium hydroxide, calcium oxide (quicklime) is then added to sodium carbonate slurry and calcium carbonate is formed, which is

termed as 'lime mud'. Trace amounts of other elements such as magnesium, potassium, sulfur, and boron can also be found in lime mud.

Lime mud is classified as a toxic industrial waste mainly due to its high alkalinity and typical mineralogical characteristics and therefore requires proper treatment before discharge [7]. Reutilization of lime mud is very limited in industries and accounts for only 30% of lime mud produced, while the remainder is disposed of in landfills. However, many serious environmental issues are associated with disposal in landfills: (a) large landfill area occupation with high disposal cost; (b) landfill leachates moving into groundwater and rivers; (c) adverse impacts on nearby plants and soils; (d) toxic fine dust generation in dry and windy conditions [8]. Furthermore, landfills are also suggested to be an unsuitable option for lime mud disposal [9]. Therefore, there is an utmost requirement for cost-effective reutilization and valorization of lime mud.

Lime mud is used in many applications, such as building materials (bricks and cements) [10], wastewater treatment [11], and agricultural soils [12]. However, high costs involved in pretreatment processes such as drying, desalting, or dewatering makes lime mud a secondary option. Therefore, lime mud valorization can be a possible option to yield a more value-added product. Lime mud is mostly composed of calcium carbonate (~95–96%), and with a few steps of pretreatment, it can be converted into calcium carbonate ($CaCO_3$), calcium oxide ($CaO$), calcium hydroxide ($Ca(OH)_2$), ceramsite, and bioceramics [13–15]. This valorization can produce $CaO$ and $Ca(OH)_2$, which has the capacity to effectively remove phosphate. The nanoscale size of produced product provides enormous surface area and activated surface for effective adsorption of phosphorus, fluoride, and many other heavy metals such as Pb, Cu, As, and Cd [16–18]. Numerous studies have been performed on bioceramics, lime sludge, and red mud (~46 wt. % of $CaO$), which have similar compositions to lime mud, for the removal of phosphorus through surface adsorption and wet chemical precipitation processes [13,19–21], whereas utilization of lime mud for phosphorus removal has not yet been documented, to the best of our knowledge.

Phosphorus is a very important element for the industries of fertilizers [22], pharmaceuticals [23], detergents [24], and batteries [25]. These industries inevitably discharge a significant amount of phosphorus-bearing wastes (mostly in the form of phosphates) in effluent streams, which ultimately causes serious environmental issues. Phosphorus is the key element responsible for the occurrence of eutrophication as it encourages the growth of algae [26,27], which then adversely affects the overall aquatic life. Severe oxygen depletion due to higher eutrophication and biological oxygen demands are some of the major consequences [28]. This directly affects the water quality and hampers aquatic life. Therefore, removal and recovery of this useful element is highly desirable to minimize any environmental impact, particularly in nearby regions of urban areas.

Conventional phosphorus treatment methods include adsorption, biological treatments, precipitation, floatation, and crystallization [29]. The reduced removal efficiencies and complex operations make most of the mentioned methods unsuitable for commercial purposes. Chemical precipitation technique is widely accepted and is also capable of removing more than 99% of phosphorus from wastewater. Moreover, precipitation of phosphorus in the form of calcium phosphate is an important physicochemical process. This further promotes the efficient and economical route for the recovery of phosphorus [28].

Numerous studies have already been conducted on the treatment of phosphorus from various chemicals in wastewater through the precipitation method [28–33]. Among them, calcium-based compounds were very effective and could even reach up to 99% phosphorus removal efficiency [32]. In particular, calcium hydroxide has some merits over other metal salts, including that it does not induce metal ions such as $Al^{3+}$ or $Fe^{3+}$ and anions such as $SO_4^{2-}$ or $Cl^-$ in the treated water. In the present work, an attempt has been made to develop green nano-calcium hydroxide from lime mud without using any toxic chemicals or energy-intensive processes.

The objectives of the present work are to valorize lime mud into a value-added product, i.e., green nano-calcium hydroxide, and to determine the optimum dosage of green nano-calcium hydroxide and residence time required for the maximum removal of phosphorus from wastewater.

## 2. Materials and Methods

### 2.1. Green Ca(OH)$_2$ Preparation

Lime mud was collected from the Moorim paper mill in Ulsan, Republic of Korea. The composition of raw lime mud was predominantly calcium carbonate with small amounts of other oxides containing $Al_2O_3$, $SiO_2$, $SO_3$, $P_2O_5$, $Na_2O$, $Fe_2O_3$, and MgO. Hydrochloric acid (HCl, 35–37% concentration) and sodium hydroxide (NaOH, 97% purity) were purchased from Junsei Chemicals, Republic of Korea. All chemicals were used as received.

The green Ca(OH)$_2$ nanoparticles were prepared by a precipitation method involving the following chemical reaction:

$$CaCO_{3 \text{ (solid)}} \text{ (lime mud)} + 2HCl_{\text{ (aqueous)}} \rightarrow CaCl_{2 \text{ (aqueous)}} + CO_{2 \text{ (gas)}} + H_2O_{\text{ (aqueous)}}$$

$$CaCl_{2 \text{ (aqueous)}} + 2NaOH_{\text{ (aqueous)}} \rightarrow Ca(OH)_{2 \text{ (solid)}} + 2NaCl_{\text{ (aqueous)}}$$

Firstly, the lime mud was ground and screened through a sieve of 100 μm. Then, 12.5 g of fine powder lime mud was dissolved in 250 mL of 1 M HCl under vigorous stirring. In order to remove residues, the solution was filtered to 0.45 μm particle size by paper filter and syringe filter. After filtering, a transparent solution was obtained. Then, the transparent solution was heated at 90 °C due to the minimum solubility of $CO_2$ in water. Moreover, the calcium hydroxide nanoparticles (CNPs) should have a perfect shape if the temperature is around 90 °C [34,35]. When the temperature of the transparent solution reached 90 °C, 200 mL of 1 M NaOH solution was added dropwise into the resultant solution under vigorous magnetic stirring while keeping the temperature of the mixed solution around 90 °C. Subsequently, the transparent mixture became white in color within 5 min. Finally, the white solution was filtered, and then the residue was washed by deionized water several times to remove remaining impurities and dried in an oven at 100 °C for one day.

### 2.2. Phosphorus Removal Study

The potassium dihydrogen phosphate ($KH_2PO_4$) (extra pure reagent, Daejung Chemicals & Metal Co., Ltd, Korea) was dissolved in DI water to prepare 15 mg L$^{-1}$ phosphorus solution. Different mass ratios of Ca(OH)$_2$ NPs/phosphorus (CNPs/P) (2.2, 3.5, 4.4, 5.3, 6.2) were added into different beakers containing 150 mL of phosphorus solution with fixed concentration of 15 mg L$^{-1}$ (pH 7) under 300 rpm magnetic stirring at room temperature and with difference contact times of 1, 5, 10, 20, and 60 min. A slight change in pH (~0.5) was observed upon addition of CNPs. The solution in the treatment beaker was directly filtered through a 0.45 μm syringe filter to separate the nanoparticles from the phosphorus solution. The filtrate was then analyzed for remaining concentration of phosphorus in the solution by a spectrophotometer (HS 3300, HUMAS) using the ascorbic acid method (3000 TP-L program). The phosphorus removal efficiency E% was determined using the following equation:

$$E\% = \frac{C_0 - C}{C_0} \times 100 \tag{1}$$

where $C_0$ is the primary concentration of total phosphorus (mg L$^{-1}$) and $C$ is the concentration of total phosphorus after treatment (mg L$^{-1}$).

### 2.3. Physical Characterization

The crystal structure study and the identification for mineral phases of lime mud waste and the prepared Ca(OH)$_2$ sample were examined by X-ray diffraction (XRD; BD2745N) with an X-ray

source of Cu K$\alpha$ ($\lambda$ = 0.15406 nm) in the scan range of diffraction angle 2$\theta$ from 20° to 80°. In order to identify characteristic functional groups and structure properties, the samples were measured by Fourier transform infrared spectroscopy (FT-IR; 6700 FTIR) with the scan range from 400 to 4000 cm$^{-1}$. The samples were directly analyzed by FT-IR without any further sample preparation. The morphological features of lime mud sample were investigated by scanning electron microscopy (SEM; JSM-6380). The Ca(OH)$_2$ NPs micro images were recorded by field emission scanning electron microscopy (FE-SEM; Hitachi-S-4800) to find out the morphologies and size of CNPs.

For analysis concentration of phosphorus before and after treatment, the solutions were measured by the water analyzer and spectrophotometer (HUMAS, HS-3300) at 880 nm using the ascorbic acid colorimetric method.

## 3. Results

### 3.1. Characteristics of Green Ca(OH)$_2$

Regarding the characterization of crystal structure of the raw lime mud and CNPs, the samples were examined by XRD, as shown in Figure 1. The raw lime mud (black line) is highly crystalline. The resultant peaks are in good agreement with the mineralogical phase of rhombohedral calcite (CaCO$_3$), having space group R-3c (space group No. 167. PDF card No. 00_081_2027). The XRD peaks at 2$\theta$ = 23.1°, 29.3°, 35.9°, 39.4°, 43.1°, 47.1°, 47.5°, 48.4°, 56.5°, 57.3°, 60.6°, and 64.6° were assigned to the (012), (104), (110), (113), (202), (024), (018), (116), (211), (122), (214), and (300) planes of the calcite phase, respectively. For Ca(OH)$_2$ nanoparticles (red line), the major diffraction peaks matched very well with the characteristic peaks of the hexagonal portlandite phase of space group P-3m1 (Space Group No. 164, PDF Card No. 00-087-0673) [34]. The peaks at 2$\theta$ = 28.6°, 34.0°, 47.0°, 50.6°, 54.2°, 62.5°, 64.1°, and 71.7° corresponded to the (100), (101), (102), (110), (111), (021), (013), and (002) planes of the portlandite phase, respectively. Besides, a minor calcite peak was found at 2$\theta$ = 29.3° due to reaction of portlandite and carbon dioxide from air. Moreover, the crystallite size (d) of produced Ca(OH)$_2$ NP powder was calculated based on the Debye–Scherrer's equation:

$$d = \frac{K\lambda}{\beta \, \cos\theta} \tag{2}$$

where $K$ is Debye–Scherrer's constant, equal to 0.090, $\lambda$ is the wavelength of X-ray radiation used ($\lambda$ = 0.15406 nm), $\theta$ is the Bragg diffraction angle, and $\beta$ is the half-width diffraction peak. The estimated mean crystal size of portlandite NPs for the plane (101) (with the highest diffraction peak) was approximately 24.8 nm. The lattice strain $\varepsilon$ of crystal at the plane was calculated with the following equation:

$$\varepsilon = \frac{\beta}{4tan\theta} \tag{3}$$

The lattice strain was found to be 4.775 × 10$^{-3}$ for the plane (101) [36–38].

To observe the structural transformation from lime mud to portlandite nanoparticles, SEM and Field Emission Scanning Electron Microscopy (FESEM) analyses were conducted. Figure 2a,b shows the scanning electron microscope images of a lime mud sample. The micrograph shows that the lime mud has irregular shape and agglomerate units. The size range of lime mud is from 300 nm to 10 $\mu$m. After preparation, the Ca(OH)$_2$ NP structure revealed irregular shape to hexagonal shape features with varying sizes, as shown in Figure 2c. Figure 2d clearly shows the hexagonal nanoplate of Ca(OH)$_2$ nanoparticles with size of approximately 550–700 nm. Furthermore, the impurity in the form of calcite in the prepared Ca(OH)$_2$ nanoparticles sample was also observed through the FESEM image. The calcite CaCO$_3$ revealed a nano-fibrous shape and nano-needle shape (Figure 2c). Other studies also verified the formation of nano-portlandite with similar morphologies and size. Darroudi et al. reported the preparation of calcium hydroxide nanoparticles with hexagonal structure and average particle size range of 600–650 nm by a facile sol–gel method in aqueous gelatin media [39]. Pereira et al.

introduced Ca(OH)$_2$ nanoplates with average dimensions of 100–300 nm and hexagonal shape [40]. Madrid et al. produced high-purity hexagonal nano-calcium hydroxide with 200–600 nm size by a precipitation method in N$_2$ atmosphere and at different temperatures [35]. Ca(OH)$_2$ nanoplates of size ranging from 350 nm to 450 nm were also developed from waste oyster shells by Khan et al. [34].

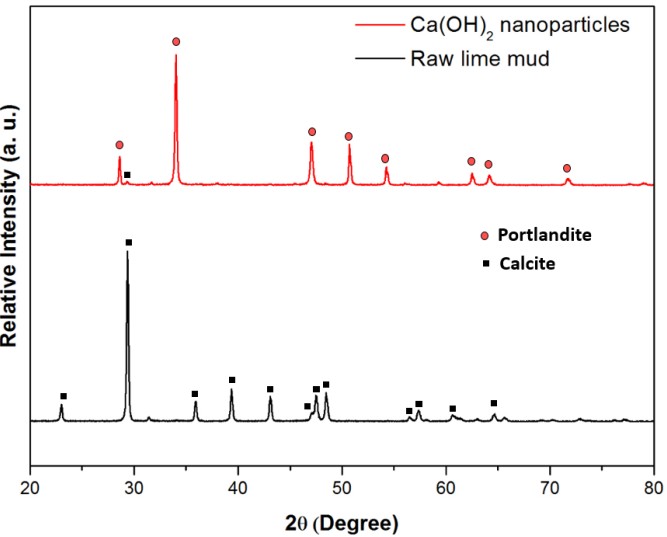

**Figure 1.** The XRD patterns of raw lime mud (black line) and Ca(OH)$_2$ nanoparticles (red line).

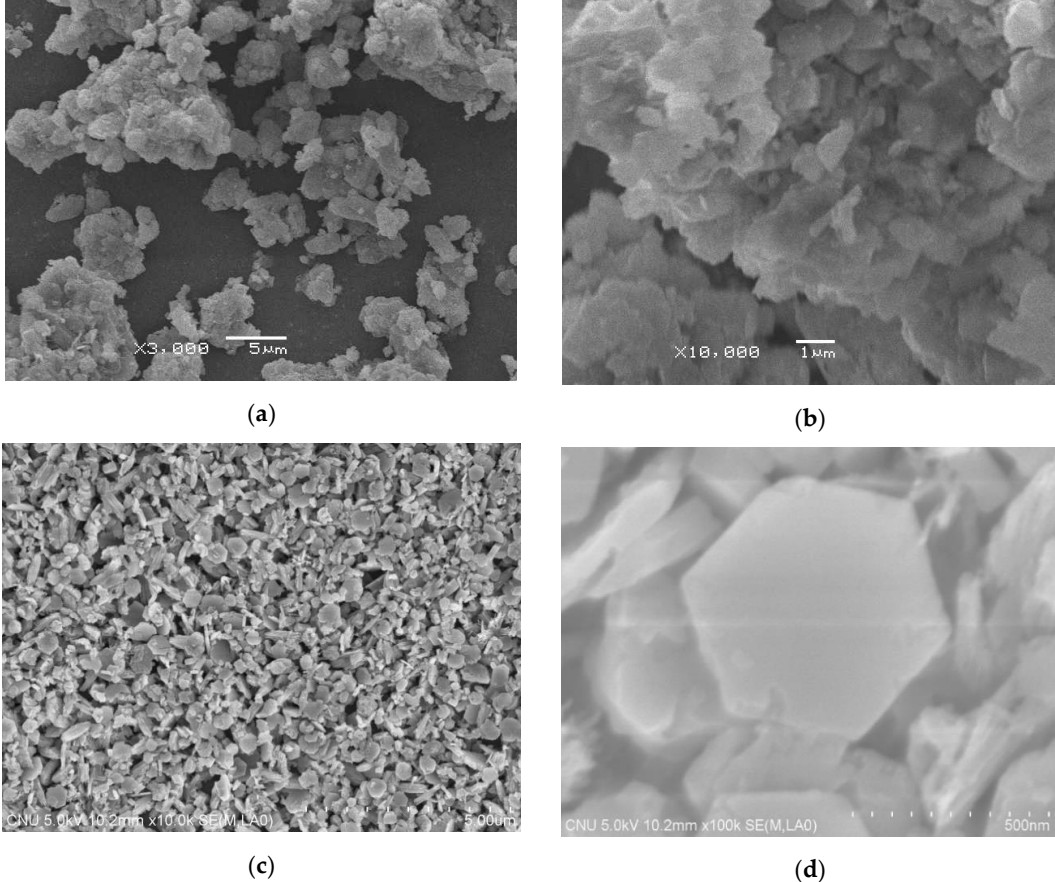

**Figure 2.** SEM analysis of lime mud samples and FESEM analysis of calcium hydroxide nano particles. (**a**) Low- and (**b**) high-magnification SEM images of raw lime mud; (**c**) low- and (**d**) high- magnification Field Emission Scanning Electron Microscopy (FESEM) of Ca(OH)$_2$ nanoparticles.

To identify the characteristic functional groups, the FTIR spectra of samples were recorded. Figure 3 presents the FTIR patterns of (a) raw lime mud and (b) $Ca(OH)_2$ nanoparticles. The FTIR spectra of raw lime mud indicated that the major composition of lime mud is calcite, as shown in Figure 3a. The broad stretching absorption and sharp peaks at 712, 871, and 1426 $cm^{-1}$ denote $v_4$ (in-plane bending mode), $v_2$ (out-of-plane bending mode), and $v_3$ (antisymmetric stretching mode) of the $CO_3^{2-}$ group of the calcite [41]. For green portlandite, the absorption peak at 3641.7 $cm^{-1}$ was assigned to the characteristic hydroxyl group ($OH^-$) stretching vibration in the portlandite phase, as shown in Figure 3b [42]. Furthermore, the weak absorption bands at 871 $cm^{-1}$ ($v_2$) and 1426 $cm^{-1}$ ($v_3$) correspond to the stretching of the carbonate ($CO_3^{2-}$) group of the calcite phase, which was also confirmed in XRD result.

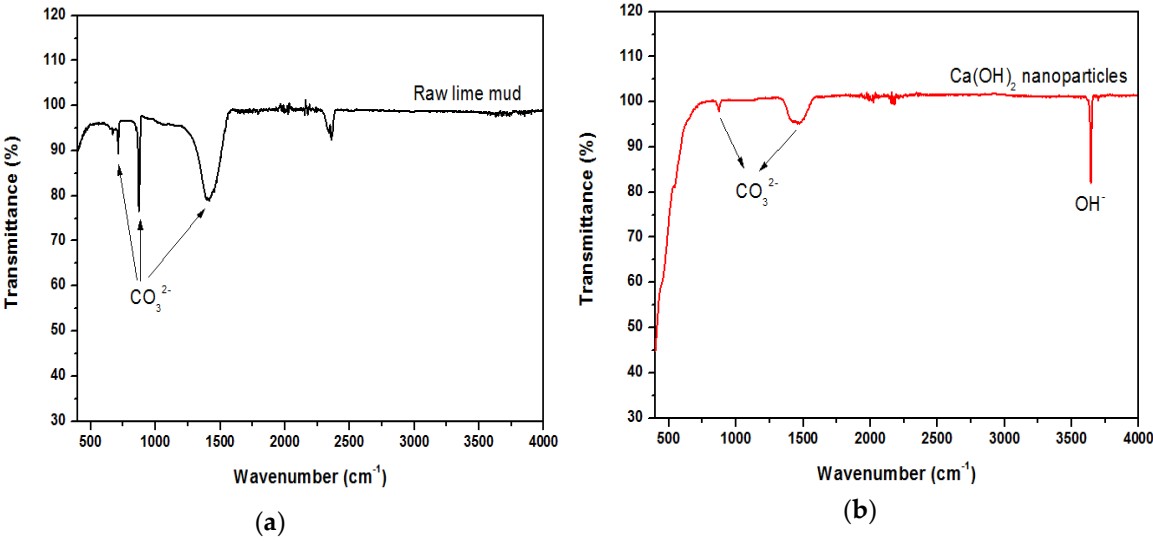

**Figure 3.** The FTIR patterns of (**a**) raw lime mud and (**b**) $Ca(OH)_2$ nanoparticles.

*3.2. Phosphorus Treatment Application*

The effect of contact time on removal of phosphorus was investigated. The experimental data were collected within 60 min to reach chemical equilibrium in the solution. The initial phosphorus concentration was fixed at 15 mg $L^{-1}$. The phosphorus removal efficiency of green calcium hydroxide versus contact time is presented in Figure 4. Initially, within 10 min, the phosphorus removal rate increased rapidly and then reduced gradually until equilibrium was attained. The highest removal efficiency of phosphorus was obtained within 20 min, as the system approaches equilibrium. The adsorption rate was rapid within the initial 10 min due to the presence of active sites on the surface of CNPs. However, due to continuous adsorption, those vacant sites started obtaining a saturation state, leading to limited contact between the phosphorus ions and surface area of the absorbent. Therefore, the phosphorus removal rate was lowered. The reaction reached chemical equilibrium when the surface of the CNPs was fully filled out by the phosphorus ions [43].

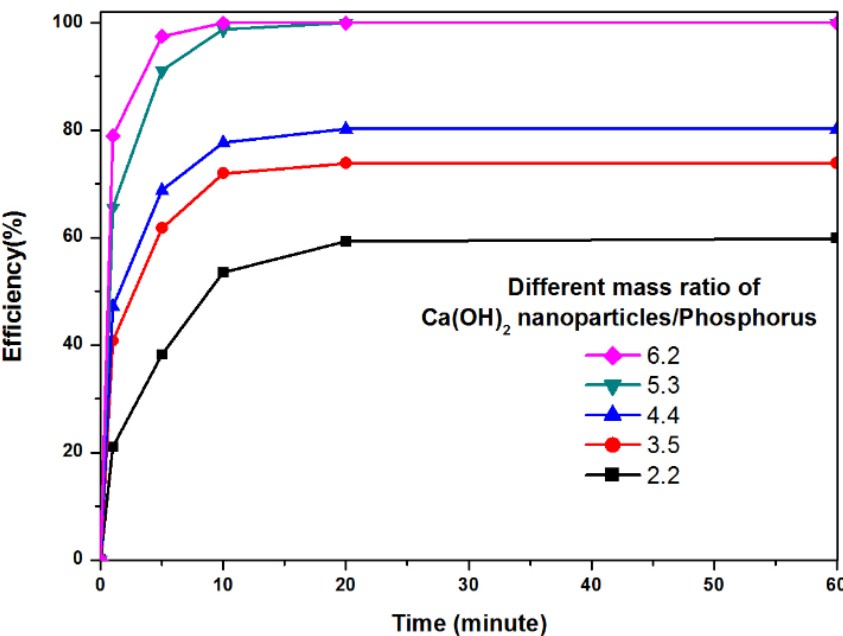

**Figure 4.** Effect of reaction time on phosphorus removal efficiency.

Figure 5 shows the effect of Ca(OH)$_2$ NP dosage on phosphorus removal within 10 min. The results show that the efficiency of phosphorus removal increased with the increased mass ratio of adsorbent green calcium hydroxide nanoparticles/phosphorus due to more vacant sites and surface area of the adsorbent particles for adsorbing phosphorus ions. When the mass ratio of CNPs/P was increased from 2.2 to 6.2, the phosphorus removal efficiency also increased from 53.5% to 100% within 10 min.

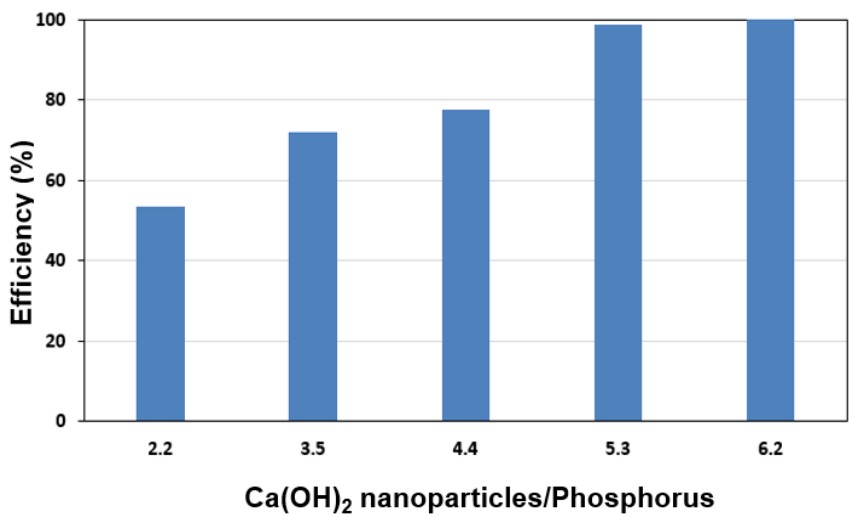

**Figure 5.** Effect of Ca(OH)$_2$ dosage (mass ratio) on phosphorus removal within 10 min.

Figure 6 illustrates the effect of absorbent dosage on removal of phosphorus with different contact time. The results indicated that the percent of phosphate adsorption increased when the mass ratio of CNPs/P and contact time were increased. When the mass ratio of CNPs/P were 2.2, 3.5, 4.4, 5.3, and 6.2, the phosphorus removal efficiencies were 59.2%, 73.8%, 80.2%, 100%, and 100%, respectively (t = 20 min). Other previous studies have also confirmed the enhancement of phosphorus removal efficiency with increasing adsorbent dosage and contact time. Torit et al. reported 80% phosphorus removal from domestic wastewater within 2 h with 5 g of calcinated eggshell in 1.7 mg L$^{-1}$ of primary phosphorus concentration, which means the mass ratio of eggshell ash/phosphorus was

about 2941 [43]. Deng et al. introduced a mass ratio of recycled concrete aggregate/phosphorus of 100, and about 95% phosphorus removal efficiency with 24 h contact time was achieved [44]. Nawar et al. determined that the removal efficiency of phosphorus ions from drinking water was 97% with a mass ratio of absorbent doses, alum sludge/phosphorus, of 50 within 30 min [45]. In this study, we reported 100% phosphorus removal efficiency with a mass ratio of CNPs/P of only 6.2 within 10 min.

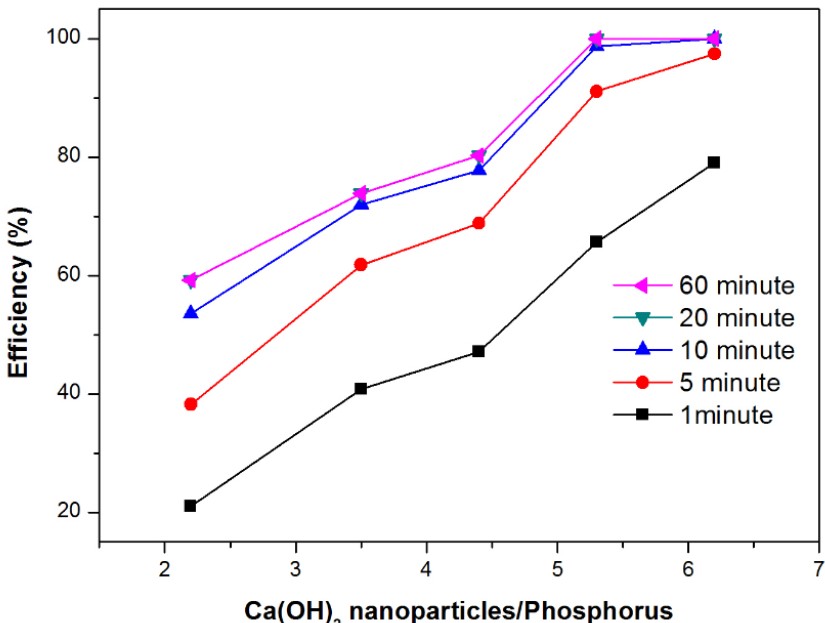

**Figure 6.** Effect of Ca(OH)$_2$ dosage on removal of phosphorus with different contact time.

Figure 7 presents the phosphorus adsorption rate and shows that phosphorus removal capacity increased until chemical equilibrium. The graph shows that the best effective adsorption capacity was achieved at a mass ratio of 6.2 of Ca(OH)$_2$ nanoparticles/phosphorus. The adsorption rate reached the maximum (adsorption capacity at equilibrium = 160.7 mg/g, efficiency = 100%) within 10 min.

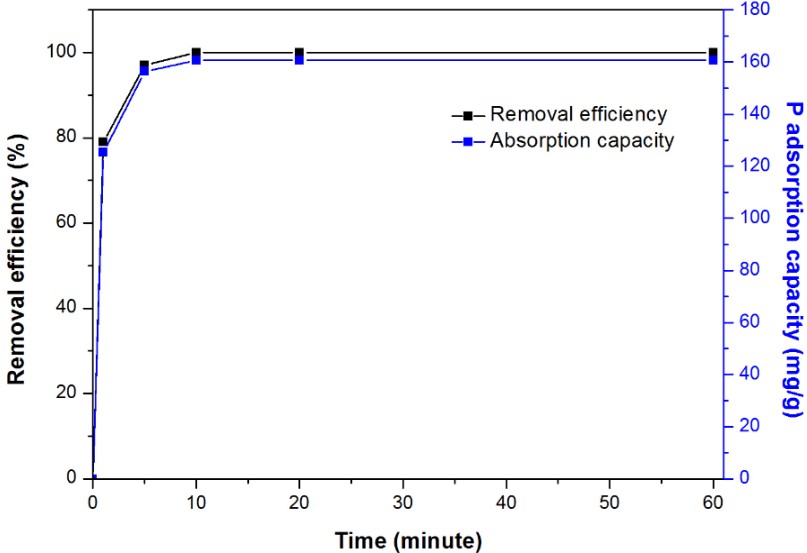

**Figure 7.** Kinetic studies of phosphorus adsorption.

## 4. Conclusions

In this work, green hexagonal portlandite NPs were successfully synthesized from waste lime mud by a precipitation method. The crystal structure and hexagonal shape of calcium hydroxide nanoparticles were determined through XRD and FESEM. The average particle size was found to be about 550–700 nm. In addition, the prepared $Ca(OH)_2$ nanoparticles were introduced as an absorbent able to remove phosphorus. A perfect phosphorus removal efficiency of 100% was achieved within 10 min when a CNPs/P mass ratio of 6.2 was employed to remove phosphorus in 150 mL of 15 mg $L^{-1}$ phosphorus solution. Due to the efficient phosphorus removal results, the nano-portlandite (CNPs) is strongly suggested as an appropriate absorbent in phosphorus treatment from domestic and industrial wastewater. Moreover, green portlandite production from lime mud has tremendous environmental and economic benefits with reduced residue storage.

**Author Contributions:** H.H.T.V., M.D.K., and R.C. planned and designed the experiment; H.H.T.V., T.Q.L., and C.R. carried out the experiments; H.H.T.V. and M.D.K. analyzed the data and wrote the paper; T.T., D.U.P., J.W.A., and J.K. reviewed and revised the manuscript.

**Funding:** This research was supported by the National Strategic Project—Carbon Mineralization Flagship Center of the National Research Foundation of Korea (NRF) funded by the Ministry of Science and ICT (MSIT), the Ministry of Environment (ME), and the Ministry of Trade, Industry, and Energy (MOTIE) (2017M3D8A2084752).

**Conflicts of Interest:** The authors declare no conflict of interest.

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
