# Peer review of "Utilization of Lime Mud Waste from Paper Mills for Efficient Phosphorus Removal"

_sustainability, doi:10.3390/su11061524_

Round 1

Reviewer 1 Report

This study presented that paper mills waste could used for phosphorus removal. This work provides a potential method to dispose toxic waste which will make the industrial more sustainable. However, there has some major issue:

1.     The study tested the waste by different dose amount and time. It is very necessary to use mathematics model to determine the maximum adsorption capacity and also to compare with other similar study.

Reference:

Deng, Y.; Wheatley, A. Mechanisms of Phosphorus Removal by Recycled Crushed Concrete. Int. J. Environ. Res. Public Health 201815, 357.

Penn, C.J.; Bryant, R.B.; Callahan, M.P.; McGrath, J.M. Use of Industrial By-products to Sorb and Retain Phosphorus. Commun. Soil Sci. Plant Anal. 2011, 42, 633–644.

2.     The introduction part needs to introduce more information about current study about lime mud and with removal mechanisms.

3.      The reference style need to meet with journal standard and the language need to improve.

Author Response

Thank you for your comments! We detailly answer your comment in the attached file.

Please kindly check the attached file. Thank you again.

Reviewer 2 Report

Review on

„Utilization of Lime Mud Waste from Paper Mills for Efficient Phosphorus Removal”

Vu et al synthesized calcium hydroxide nanoparticles (CNP) from waste material of paper mills in order to test their capability for phosphorus removal. Several short precipitation experiments were conducted testing the effect of reaction time and CNP amount on the phosphorus removal efficiency. Furthermore, different properties of the synthesized CNP material were analyzed by using three well-established structure-analytical methods. Overall, this is an interesting valuable work and the rationale becomes nicely explained in the introduction. However, there are a large number of linguistic flaws (word order, word usage, missing articles and words) and in some cases some clarification or more information is needed also to enable better comparison of this work with other studies. Since English is also not my mother tongue my minor comments regarding linguistic errors or awkward sentences might be not comprehensive in following. Therefore, I strongly recommend a final professional text editing after careful revision of the manuscript. There are few more major issues I would recommend to re-assess:

1)    I clearly see the advantages to synthesis nanoparticles from a waste product but is it possible to exclude any contamination which produce later side effects while it becomes used as “filter material”.

2)    I would also like to know if this synthesized material was tested beforehand in other studies for its phosphorus removal capability. In this context it would be nice to get some more information about the advantages and limitations of this tested product compared to other compounds.

3)    The rationale of making these different analytical efforts using field emission scanning electron microscopy, Fourier transform infrared spectroscopy and X-ray diffraction should be better explained already in the method section since not all readers might be familiar with the applied techniques.

Minor comments

Line 18: I would suggest to read: “The NP’s also termed green portlandite with ….”

Line 21: The abbreviations (FE-SEM, FT-IR, XRD) in the abstract can be omitted since not further used here. In the text it must be introduced again anyway.

Line 25: Explain CNP’s

Line 25: This information is not useful here. Addition to which volume of P solution having 15 mg P/L? I would suggest to indicate mass ratios or if possible better molar ratios NP/P so that it becomes clear and comparable with other precipitating materials.

Lines 25-26: omit “results” (“Due to the efficient phosphorus removal, the ….”

Line 58. Missing word, possibly read “can be a possible option as more value-added product”

Line 64: I assume “eutrophication” is meant instead of “acidification”

Line 65: replace “make it unsuitable for aquatic life” with “impair” or “hamper aquatic life”

Line 99: “°C” check also rest of the text

Line 100: introduce CNP and use abbreviation afterwards

Line 111: what was the pH after adding CNP?

Lines 118-119: I assume that soluble reactive phosphorus was analysed subsequent to filtration of water samples (which kind of filters? Pore size 0.45 µM?)  and without any digestion of the samples? If so another term than total phosphorus should be used. Where the samples after adding CNP centrifuged before of filtration?

Lines 123-125: What was the sample preparation of the raw lime mud for FT-IR analysis. Was it measured using the transmission technique?

Line 181: “hydroxyl group”

Lines 186-198: This part is rather poorly written, in particular the usage of some terms is not appropriate : “optimal adsorption time”, “to full equilibrium”, and “optimal removal efficiency”. Alternatively, I would suggest “reaction time”, “to reach chemical equilibrium”, “highest removal efficiency”. The whole section should be carefully rewritten, since there are some more text problems like “The phenomenon suggests that the rapid adsorption rates revealed within initial 10 min because the ions of phosphorus were adsorbed rapidly on the active surface of calcium hydroxide particles.” But also the following sentences are rather awkward.

Lines 204-214: I would suggest to translate the dosage information into molar or mass ratios of the reactants which is more reasonable.

Line 205: I would suggest “phosphorus removal efficiency”

Lines 215-217: Sentence must be rewritten.

Lines 217-222. Again, all of the information presented here is not very useful as long as the mass/molar ratios of the reactants are not indicated. The usage of concentration unit is also not consistent.

Line 226-234: The section must be rewritten as well! As mentioned before, please translate the dosage information in a more meaningful way. This section should only include what was investigated/examined in this study! I cannot find any information about the cost efficiency (even this is indeed an interesting aspect). Consider also deleting the last sentence (not investigated in this study). Line 228: use another word than “aqueous”, line 230 use another word than “excellent”

Figures: please ensure that all Figures are self-explaining (without reading the text), all abbreviations must be clarified. For Figure 4 I suggest rewording: “Effect of reaction time on phosphorus removal efficiency”, Figure 5, 6: the dosage information in both figures  is meaningless, better use mass/molar ratios as suggested before.  

Author Response

(The authors gave the same response as above.)

Round 2

Reviewer 1 Report

The paper met with most of requirements. Recommend to double check some small mistake.

Author Response

Response to Reviewer 1 Comments

Comments

The paper met with most of requirements. Recommend to double check some small mistake.

Ans) Thank you very much for your comments. We checked carefully the manuscript and corrected all mistake

Reviewer 2 Report

Dear authors,

the manuscript is indeed nicely revised. Just few little points. 

1) The first might be just a misunderstanding. Now in the abstract only the abbreviations are used for the applied methods (FE-SEM, SEM, ...). Actually you should write the full words here and the abbreviations can be omitted in the abstract! So I would recommend changing it (the other way around you did) even most readers might be familiar with terms.

2) In figure 4 I would suggest to turn the legend showing the mass ratios: lowest rate at the bottom and highest at the top, the same for Figure 6

3) consider some changes (red colored) for figure 5:  "Effect of Ca(OH)2 dosage (mass ratio) on phosphorus removal within 10 minutes."

4) Line 225 replace "phosphorus ion" with "phosphate"

5) Line 234: please round value "97%"

Author Response

Response to Reviewer 2 Comments

Comments

The manuscript is indeed nicely revised. Just few little points. 

Ans) Thank you very much for your encouragement and suggestions. We revised the manuscript as per your suggestions.

The first might be just a misunderstanding. Now in the abstract only the abbreviations are used for the applied methods (FE-SEM, SEM, ...). Actually you should write the full words here and the abbreviations can be omitted in the abstract! So I would recommend changing it (the other way around you did) even most readers might be familiar with terms.

Ans) Thank you for your comment. We modified as your recommendations in line 23-24.

 In figure 4 I would suggest to turn the legend showing the mass ratios: lowest rate at the bottom and highest at the top, the same for Figure 6

Ans) Thank you for your comment. As per your recommendations, modified the graphs in the revised manuscript.

Consider some changes (red colored) for figure 5:  "Effect of Ca(OH)2 dosage (mass ratio) on phosphorus removal within 10 minutes."

Ans) We have modified as “Effect of Ca(OH)2 dosage (mass ratio) on phosphorus removal within 10 minutes.”

Line 225 replace "phosphorus ion" with "phosphate"

Ans) We replaced “phosphorus ion” by “phosphate” in line 225

Line 234: please round value "97%"

Ans) We corrected the 97% in line 234
